# Cavitating Jet: A Review

**Hitoshi Soyama**

Department of Finemechanics, Tohoku University, Sendai 980-8579, Japan; soyama@mm.mech.tohoku.ac.jp;
Tel.: +81-22-795-6891; Fax: +81-22-795-3758

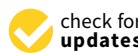

**Featured Application: Cavitation peening, Cleaning, Drilling.**

**Abstract:** When a high-speed water jet is injected into water through a nozzle, cavitation is generated in the nozzle and/or shear layer around the jet. A jet with cavitation is called a "cavitating jet". When the cavitating jet is injected into a surface, cavitation is collapsed, producing impacts. Although cavitation impacts are harmful to hydraulic machinery, impacts produced by cavitating jets are utilized for cleaning, drilling and cavitation peening, which is a mechanical surface treatment to improve the fatigue strength of metallic materials in the same way as shot peening. When a cavitating jet is optimized, the peening intensity of the cavitating jet is larger than that of water jet peening, in which water column impacts are used. In order to optimize the cavitating jet, an understanding of the instabilities of the cavitating jet is required. In the present review, the unsteady behavior of vortex cavitation is visualized, and key parameters such as injection pressure, cavitation number and sound velocity in cavitating flow field are discussed, then the estimation methods of the aggressive intensity of the jet are summarized.

**Keywords:** cavitation; jet; vortex; mechanical surface treatment; cavitation peening

## 1. Introduction

Cavitation is a harmful phenomenon for hydraulic machineries such as pumps, as severe impacts are produced at bubble collapse [1,2]. However, cavitation impacts are utilized for mechanical surface treatment in the same way as shot peening, and this is named "cavitation peening" [3,4]. The great advantage of cavitation peening is that shots are not used in the peening process, as cavitation impacts are used in cavitation peening [5]. Thus, the cavitation-peened surface is less rough compared with the shot-peened surface, and the fatigue strength of cavitation peening is better than that of shot-peening [6]. In conventional cavitation peening, cavitation is generated by injecting high-speed water jet into water [3,4], and a submerged water jet with cavitation is called a "cavitating jet". The cavitation peening is utilized for the impacts of cavitation collapses, and it is different from water jet peening, in which water column impacts are used. To use the cavitating jet for peening, it is worth understanding the mechanism of the cavitating jet.

In a cavitating jet, cavitation is produced inside and/or outside a nozzle when sufficient pressure difference is applied across the nozzle. As Monkbadi reviewed, vortex structures in turbulent jets [7], ring vortices (0-mode), a single helical vortex (1st mode) and double-helical vortices (2nd mode) were observed in a cavitating jet [8–10]. In the case of a developed cavitating jet, which has been used for practical applications such as cutting, material testing, drilling and peening [11–17], cavitation clouds are shed periodically [16,18–27]. As it was reported that the lifetime of the cloud is a key factor in the aggressive intensity of the cavitating jet [28], the investigation of cavitation cloud shedding is very important. On the other hand, from the viewpoint of cavitation inception, a turbulent jet with cavitation was investigated [29–32]. In the present review, in order for the cavitating jet to be used

for practical applications, the main subject is the developed cavitating jet, as shown in Figure 1a. In Figure 1, white bubbles are cavitation bubbles, as a used flash lamp was placed at the same side of a camera. As shown in Figure 1a, the cavitation clouds are clearly observed in the cavitating jet in water.

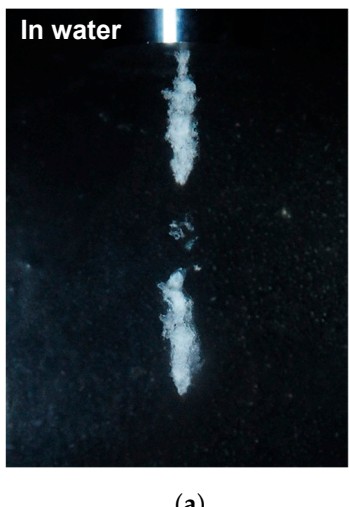

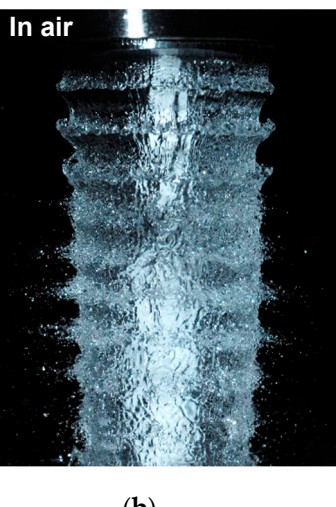

(**a**)           (**b**)

**Figure 1.** Typical aspects of cavitating jet: (**a**) cavitating jet in water; (**b**) cavitating jet in air.

Normally, a cavitating jet is produced by injecting a high-speed water jet into a water-filled chamber, as mentioned above. To apply a cavitating jet for components and/or plants which cannot be put in the chamber, Soyama developed a cavitating jet in air by injecting a high-speed water jet into a low-speed water jet, which was injected into air without the water-filled chamber using a concentric nozzle [33–36]. A typical cavitating jet in air by injecting a high-speed water jet into a low-speed water jet is shown in Figure 1b. Cavitation clouds are also observed in the water column of low-speed water jets of cavitating jets in air, as shown in Figure 1b. Even though the cavitating jet was in air, cloud shedding was observed [35,37,38]. In addition, in the case of the cavitating jet in air at optimum conditions, the wavy pattern of the low-speed water jet was observed, as shown in Figure 1b [34,35,37,38]. The frequency of the wavy pattern is equal to the shedding frequency of the cloud [35]. Even for the cavitating jet in air, unsteady behavior is very important.

In cavitating jets both in water and in air, whereas the cloud shedding frequency is several hundred Hz [18,35,39,40], only a few severe impacts occur per second [34,41–43] when cavitation impacts are measured by special-made PVDF transducers [41,42]. Thus, in order to enhance the aggressive intensity of the cavitating jet in water and air for practical applications of cavitating jets, the unsteady behavior of the cavitating jet should be investigated. In the present review, the normal cavitating jet, i.e., the cavitating jet in water, was mainly discussed, and the cavitating jet in water was simply described as the cavitating jet.

To simulate the cavitating jet numerically, it is important to understand the flow pattern of the cavitating jet. As is well known, numerical simulation of cavitating flow is not yet easy, due to the high Reynolds number and phase change phenomenon. Numerical investigation of three-dimensional cloud cavitation with special emphasis on collapse induced shock dynamics [44], simulation of cloud cavitation on propeller [45], and the hydrofoil [46] were carried out. In the area of numerical simulation of the cavitating jet, considering bubble dynamics, the numerical simulation of vortex cavitation in a three-dimensional submerged transitional jet near inception condition was carried out [47], and bubble growth in the shear layer of the cavitating jet was calculated [48]; residual stresses introduced by cavitating jet considering bubble growth and collapse were also simulated [49]. The cavitating flow in a venturi nozzle was also tried by a large eddy simulation of turbulence–cavitation interactions [50]. From the viewpoint of cloud-shedding of the cavitating jet, numerical analysis of cavitation cloud-shedding in a free submerged water jet was carried out [51,52].

The interaction of cavitation bubbles and materials was also investigated numerically [53–56]. In the case of the aggressive cavitating jet, the clouds shed periodically and form ring vortex cavitation on the impinging surface, and then collapse, producing severe impacts. Thus, the experimental investigation of the flow pattern would be assisted by numerical simulations for more precise investigations of the cavitation impacts produced by the cavitating jet.

In the present review, in order to use the cavitating jet for practical applications, papers that investigated the flow pattern of the cavitating jet experimentally were reviewed. To enhance the aggressive intensity of the cavitating jet for these applications, the key factors of the cavitating jet were also summarized, and the estimation method of the aggressive intensity of the jet was discussed. In the present paper, the aggressive intensity of the cavitating jet means erosion rate measured by weight loss of the target metals and/or the peening intensity measured by the arc height of the metallic plate.

## 2. Cavitation

Cavitation is a phase-change phenomenon in which the liquid phase is changed to a gas phase due to a decrease in liquid pressure to vapor pressure by increasing flow velocity [2], and it is called "hydrodynamic cavitation". Cavitation is also generated by ultrasonic vibration, which is named "ultrasonic cavitation". Bubbles are also formed by irradiating a pulse laser into water, in which the bubble behaves as a cavitation bubble [57]; this is called "laser cavitation". A schematic diagram of cavitation is illustrated in Figure 2a. When a cavitation nucleus such as a tiny air bubble is subjected to a low-pressure region, it becomes a cavitation, and it develops and shrinks, then collapses, producing microjet and shock wave during rebound. The microjet and the shock wave produce a severe impact, which can deform metals. After rebound and shrinking, bubbles remain in the water, which is called a residual bubble [58]. In the research area of cavitation, including bubble dynamics, spherical bubbles have mainly been investigated by numerical simulations [59] and experimental studies [57,60,61], and the effects of bubble shape and bubble interactions have also been studied [62].

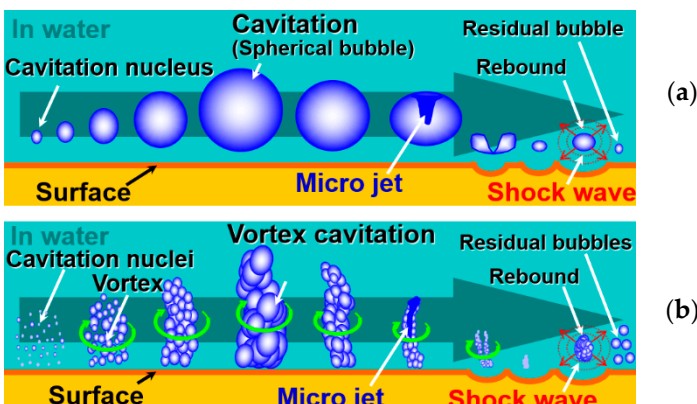

**Figure 2.** Schematic diagram of development and collapse of cavitation: (**a**) spherical bubble; (**b**) vortex cavitation.

In experimental studies of severe cavitation erosion, it was found that vortex cavitation, as shown in Figure 3 [63], produced severe impacts on fluid machineries such as pumps and valves [1,64,65]. A schematic diagram of vortex cavitation is shown in Figure 2b. Cavitation nuclei accumulate in a vortex, and they become vortex cavitation in a high-speed region, i.e., a low-pressure region. The vortex cavitation develops and shrinks, then collapses. Regarding a model test of vortex cavitation using a rotating device [66], a microjet was observed in the vortex cavitation. Thus, in order to use cavitation impacts for practical applications, the generation of vortex cavitation is very important.

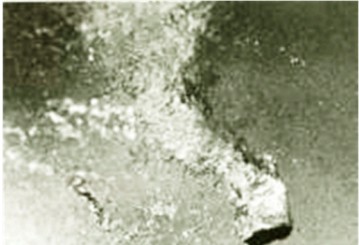

**Figure 3.** Typical aspect of vortex cavitation, which produces severe impact [63].

## 3. Cavitating Jet

### 3.1. Structure of Cavitating Jet

To show the flow pattern of the cavitating jet, Figure 4 illustrates aspects of the impinging cavitating jet taken with (a) a flash lamp of 1.1 μs and (b) a shutter speed of 1/25 sec, i.e., 40 ms. Thus, Figure 4a reveals an instantaneous aspect of the jet, and Figure 4b shows a kind of time-averaged aspect of the impinging cavitating jet. In the nozzle used for the cavitation jet, the cavitator and the guide pipe were installed to enhance the aggressive intensity of the cavitating jet [39], and standoff distance was defined as the distance from the upstream corner of the nozzle to the specimen surface. As shown in Figure 4a, a cloud cavitation was observed between the nozzle and the impinging surface, and a ring vortex cavitation was observed on the surface. When the cavitating jet was observed by a normal light, as shown in Figure 4b, the cavitating jet seems to be a continuous jet from a nozzle to an impinging surface. However, cloud cavitation sheds from the nozzle to the surface, and the cloud cavitation becomes a ring vortex cavitation.

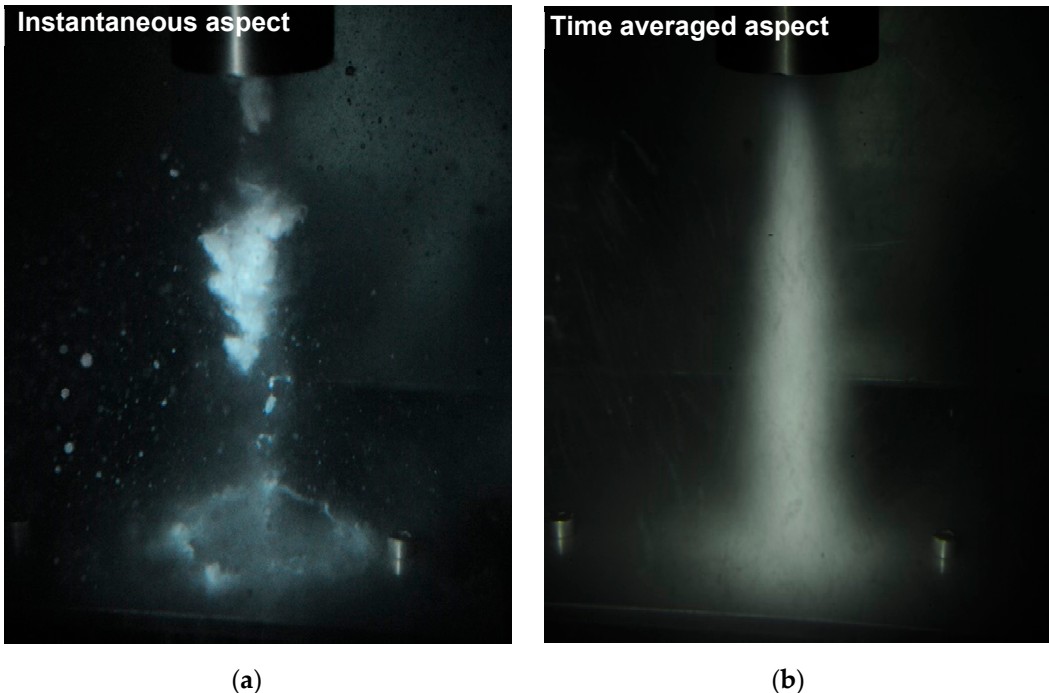

(**a**)            (**b**)

**Figure 4.** Aspects of cavitating jet (cylindrical nozzle, nozzle diameter $d$ = 2 mm, upstream pressure $p_1$ = 30 MPa, downstream pressure $p_2$ = 0.1 MPa, standoff distance $s$ = 222 mm): (**a**) observation with flash lamp whose exposure time is 1.1 μs; (**b**) observation with normal light and shutter speed of 40 ms.

To show vortex cavitation in the cavitating jet more precisely, Figure 5a reveals the aspect of the free cavitating jet through a conical nozzle, and Figure 5b shows the aspect of the impinging cavitating jet with a flash lamp [9]. As shown in Figure 5b, the PVDF transducers [41,42] were installed

in the impinging surface, which the white arrow shows, and the signal from the PVDF transducer was synchronized to the flash lamp; thus, Figure 5b reveals the aspect of the jet which produces the impact on the impinging surface. As shown in Figure 5a, helical vortex cavitation is observed near the nozzle outlet, and cloud cavitation is observed downstream from the nozzle. Thus, the cloud cavitation results from merging the vortex cavitation. In the case of the impinging cavitating jet, a part of the ring cavitation collapses on the surface producing the impact, which was detected by the PVDF transducer. The ring vortex cavitation produces a severe impact on the impinging surface. In view of the practical applications of cavitation impacts generated by the cavitating jet, the collapse of the ring vortex cavitation is an important phenomenon.

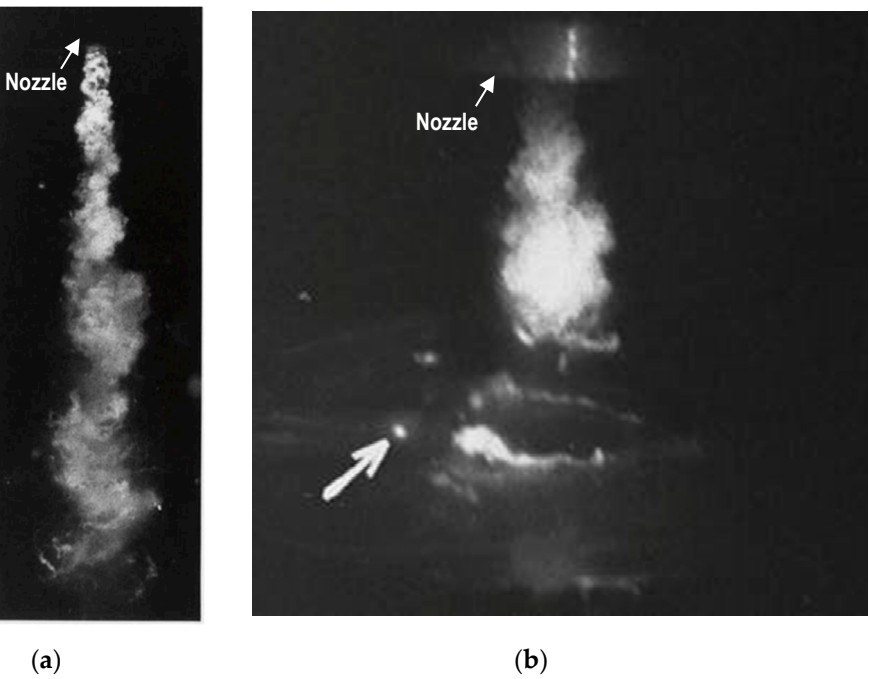

(a) (b)

**Figure 5.** Vortex cavitation in cavitating jet: (**a**) helical vortex cavitation in free cavitating jet (conical nozzle, nozzle diameter $d$ = 2.1 mm, upstream pressure $p_1$ = 6 MPa, downstream pressure $p_2$ = 0.18 MPa); (**b**) ring vortex cavitation on impinging surface (cylindrical nozzle, nozzle diameter $d$ = 2 mm, upstream pressure $p_1$ = 12 MPa, downstream pressure $p_2$ = 0.36 MPa, standoff distance $s$ = 40 mm) [9].

To reveal periodical shedding of cloud cavitation from the nozzle, as mentioned in the introduction, Figure 6 shows the aspect of the impinging cavitating jet taken by a high-speed video camera. The nozzle shown in Figure 6 was the same nozzle as in Figure 4, and it had the cavitator and the guide pipe. At $t$ = 0 ms, the cavitation cloud sheds from the nozzle to the downstream, and the cloud reaches the impinging surface at $t$ = 1.5 ms. A part of cloud cavitation becomes a ring vortex cavitation at $t$ = 1.75 ms, and it spreads out on the surface and then collapses. Although the water jet is injected continuously into water, the vortex cavitations near nozzle shed downstream coalescing each other and they become a large cloud cavitation, thus the bubble density between clouds is reduced. At $t$ = 4.25 ms, the new cavitation cloud, whose shape is similar to that at $t$ = 0 ms, sheds from the nozzle. Thus, it is a periodical phenomenon, as previously reported [16,18–27].

According to a previous report [28], as shown in Figure 7, some cloud, i.e., the 1st cloud in Figure 7a, stays near the nozzle, and the 2nd jet core passes through the 1st cloud, as shown in Figure 7b,c; then, the 2nd jet core produces the 2nd cloud downstream of the 1st cloud. The 3rd jet core passes through the 1st and 2nd clouds and produces the 3rd cloud downstream of the 2nd cloud, as shown in Figure 7d. The 5th jet core produces the 5th cloud downstream of the 1st cloud as shown in Figure 7e. Detailed images of the free cavitating jet observed by the high-speed video camera and the schematic cross-sectional diagrams of the progress of a cavitating jet are shown in

reference [28]. In Figure 6, the cavitation cloud marked by the yellow arrow stays at nearly the same position. The cloud cavitation staying where it is means that it has a long lifetime. The jet core with a longer lifetime cavitation cloud impinges the surface with high impingement pressure, as a drag of the jet core with the larger cloud is lesser than that of a smaller cloud because of the density of water and bubbles. Thus, the aggressive intensity of the cavitating jet with a longer lifetime cavitation cloud is larger.

Namely, the lifetime of the cloud is a key factor of the cavitating jet, and the longer the cloud lifetime, the more aggressive the intensity of the cavitating jet [28].

Figure 8 shows a typical aspect of a pure aluminum specimen that was exposed to the fixed cavitating jet, revealing the treatment area of the fixed cavitating jet. Plastic deformation pits are observed in a ring region, whose outer diameter is 60 mm and inner diameter is 30 mm. When the nozzle is scanned or the specimen is moved, the treatment area is uniform [67]. In Figure 8, as the nozzle throat diameter was 2 mm, the cavitating jet can treat an area 30 times wider than that of the nozzle throat. While the white region, which was impinged by the jet core, was observed at the jet center, the main treatment area is the ring region. Namely, the jet center is not treated by cavitation impacts. The understanding of the treatment area by the cavitating jet shows that a ring region is very important when using the cavitating jet for the practical applications.

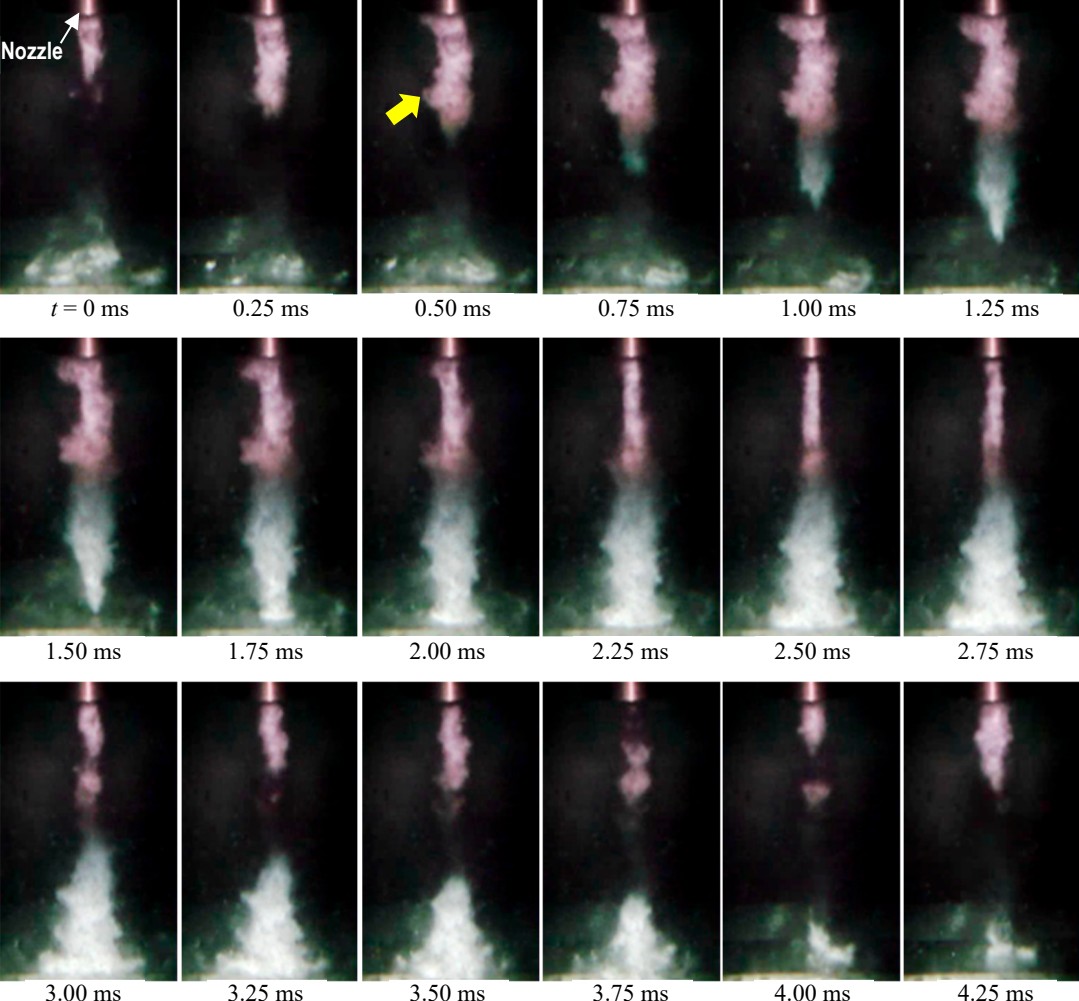

**Figure 6.** Aspects of impinging cavitating jet observed by high-speed video camera (cylindrical nozzle, nozzle diameter $d$ = 2 mm, upstream pressure $p_1$ = 30 MPa, downstream pressure $p_2$ = 0.1 MPa, standoff distance $s$ = 222 mm).

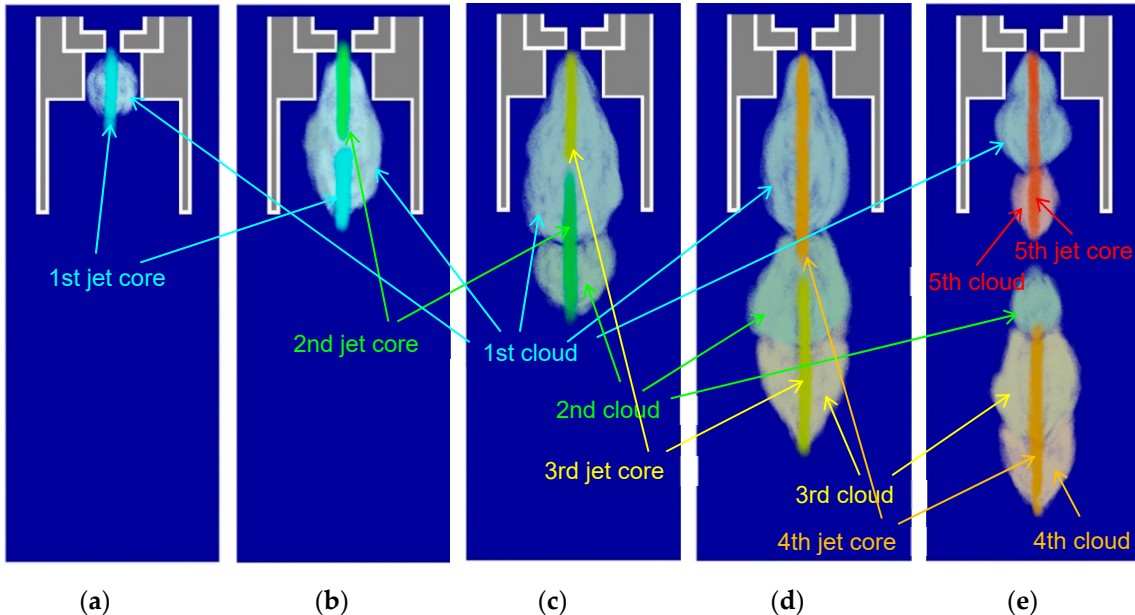

**Figure 7.** Schematic diagram of the development of cavitation cloud in cavitating jet: (**a**) 1st cloud produced by the 1st jet core; (**b**) 2nd jet core passing through the 1st cloud; (**c**) 2nd cloud produced by the 2nd jet core and 3rd jet core passing through the 1st cloud; (**d**) 3rd cloud produced by the 3rd jet core and 4th jet core passing through the 1st and 2nd clouds; (**e**) 4th cloud produced by the 4th jet core, and 5th cloud produced by 5th jet core passing through the 1st clouds.

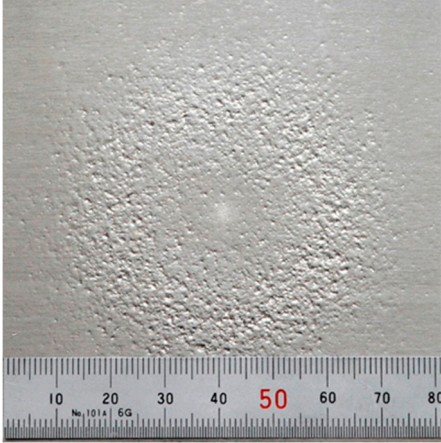

**Figure 8.** Typical treatment area by a fixed cavitating jet (pure aluminum, nozzle diameter $d$ = 2 mm, upstream pressure $p_1$ = 30 MPa, downstream pressure $p_2$ = 0.1 MPa, standoff distance $s$ = 262 mm, exposure time $t$ = 1 min) [4].

Regarding the reason for the ring region, Figure 9 illustrates the schematic diagram of the impinging cavitating jet, considering the observations of the cavitating jet by the instantaneous photograph and the high-speed video. As shown in Figure 4b, when the cavitating jet is observed by an instantaneous photograph with normal light, the cavitating region seems to be a continuous jet, as shown in Figure 9a. It was previously thought that swirl cavitation in the shear region around the jet directly hits the impinging surface and that the swirl directly produces a ring treatment area [68–71]. However, this is incorrect, because the aspect of the cavitating jet, as shown in Figure 4b is a kind of time-averaged cavitating region. Considering Figures 4a, 5 and 6, vortex cavitation is initiated in and near the nozzle outlet, and cloud cavitations combine each other; then, the cloud cavitation forms the ring vortex cavitation on the impinging surface. Thus, in order to simulate bubbles in the cavitating jet, the pressure hysteresis of these processes should be considered. For example, if the cavitation in

swirl directly hit the impinging surface, the pressure around cavitation would be the pressure in the shear layer around the jet, and it gradually increases with the shedding of cavitation, as the jet speed decreases from the distance from the nozzle. On the other hand, the pressure of the cloud cavitation, which impinges the surface, increases at the impinging jet center suddenly and decreases in the ring vortex cavitation, then increases again.

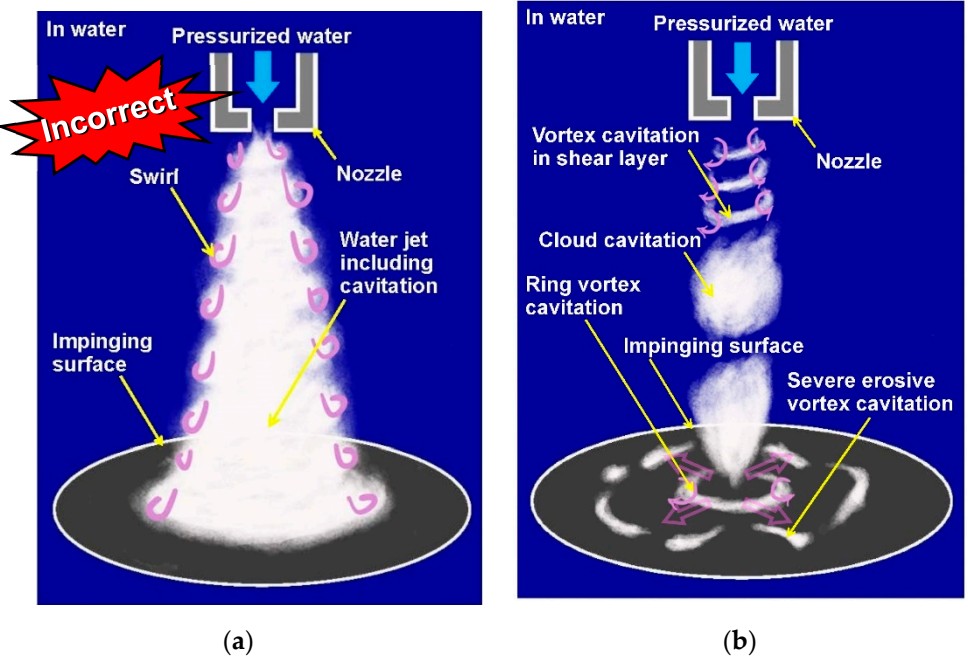

**Figure 9.** Schematic diagram of cavitating jet: (**a**) swirl; (**b**) vortex, cloud and ring cavitation.

To consider the mechanism of the ring erosion occurring on the impinging surface, Figure 10 shows a schematic diagram of the local cavitation number on the surface [72]. The impinging pressure profile on the surface has a maximum at the jet center, $p_{max}$, and it changes with the injection pressure and cavitation number. The $b$ in Figure 10 is a kind of jet width defined by the flow velocity [72]. The cavitation number of the cavitating jet, $\sigma$, is defined by the injection pressure, i.e., the upstream pressure of the nozzle $p_1$, the downstream pressure of the nozzle $p_2$ and the vapor pressure of water $p_v$ as follows [2], and is simplified as Equation (1) and as $p_1 >> p_2 >> p_v$.

$$\sigma = \frac{p_2 - p_v}{p_1 - p_2} \approx \frac{p_2}{p_1} \tag{1}$$

In Figure 10, the local cavitation number $\sigma_L$ is defined by Equation (2) and is proportional to the ratio of $p - p_v$ and $\frac{1}{2} \rho v_{max}^2$ [72].

$$\sigma_L \propto \frac{p - p_v}{\frac{1}{2} \rho v_{max}^2} \propto \frac{p - p_v}{p_{max} - p_2} f(r) \tag{2}$$

Here, $p$ and $v_{max}$ are the pressure and the maximum flow velocity on the impinging surface. The $v_{max}$ has a maximum at a certain distance from the jet center, $r$, as it is zero at the jet center and at a further point from the jet center as shown in the lower figure of Figure 10. As $v_{max}$ is mainly determined by the pressure difference, $p_{max} - p_2$, $\sigma_L$ is described by a function of $f(r)$, as shown in the right-hand term of Equation (2). When the ring vortex cavitation sheds on the surface, the cavitation develops the $\partial \sigma_L / \partial r < 0$ region, and it collapses at $\partial \sigma_L / \partial r \approx 0$. This is why the ring treatment area is obtained by the impinging cavitating jet. The detail of the pressure distribution on the impinging surface was shown in references [72,73].

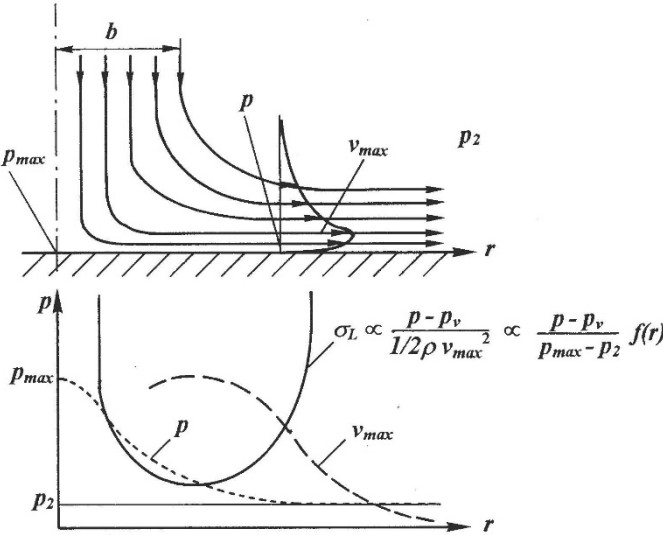

**Figure 10.** Schematic diagram of local cavitation number [72].

### 3.2. Periodical Shedding of Cavitation Cloud

As mentioned above, the cavitation cloud sheds periodically [16,18–27], and it forms the ring vortex cavitation; then, the ring vortex cavitation collapses, producing impacts. Thus, the periodical shedding of the cavitation cloud is an important phenomenon. Figure 11 reveals aspects of periodical shedding of the cavitation cloud of a cavitating jet [21]. In Figure 11, the jet flows from the left-hand side to the right-hand side. As shown in Figure 11a, the cavitation cloud breaks near the nozzle at 0.5, 0.9, 1.4 and 1.9 ms; thus, the shedding frequency $f_{shedd}$ is about 2 kHz for $p_1$ = 20 MPa and $\sigma$ = 0.014. When $p_1$ is increased to 30 MPa at a constant cavitation number, i.e., $\sigma$ = 0.014, $f_{shedd}$ is increased to 2.4 KHz, as shown in Figure 11b. When $\sigma$ is increased from 0.014 to 0.02 at constant $p_1$= 20 MPa, $f_{shedd}$ also increases from 2 to 3.2 KHz, as shown in Figure 11a,c. $f_{shedd}$ is affected by not only $p_1$ but also $\sigma$. As is well known, the cavitating length $L_{cav}$ and the width $w_{cav}$ are affected by $p_1$, $\sigma$ and $d$. In a previous report, $f_{shedd}$ was measured experimentally, changing with $p_1$, $\sigma$ and $d$, and the following relations were found [21].

$$f_{shedd} \propto w^{-1} \tag{3}$$

$$f_{shedd} \propto p_1^{0.45\pm0.03} \tag{4}$$

$$f_{shedd} \propto d^{-0.98\pm0.14} \tag{5}$$

$$f_{shedd} \propto \sigma^{0.83\pm0.10} \tag{6}$$

When the Strouhal number, $St$, is defined by $f_{sedd}$, $w$ and jet velocity at the nozzle exit, $U$, which is calculated from $p_1$, the following experimental equation is obtained [21].

$$St = \frac{f_{shedd} \cdot w}{U} \approx 0.18 \pm 0.02 \tag{7}$$

This result suggests that the cavitation cloud shedding is a phenomenon governed by a constant Strouhal number. On the other hand, the nozzle outlet geometry affects the aggressive intensity of the cavitating jet [39,74]. Details are described in Section 4.6. The $f_{sedd}$ is also affected by the nozzle outlet geometry. For example, when the guide pipe with the cavitator was installed, the aggressive intensity of the jet was four times larger than that without the guide pipe and the cavitator [39], and $St$ became nearly a quarter of that of the jet without the guide pipe and the cavitator [75], as $f_{sedd}$ was decreased. Namely, the constant value of $St$ should be unique to the nozzle outlet geometry. The investigation of $St$ for various nozzles would be a future work, to enhance and/or control the aggressive intensity of the cavitating jet.

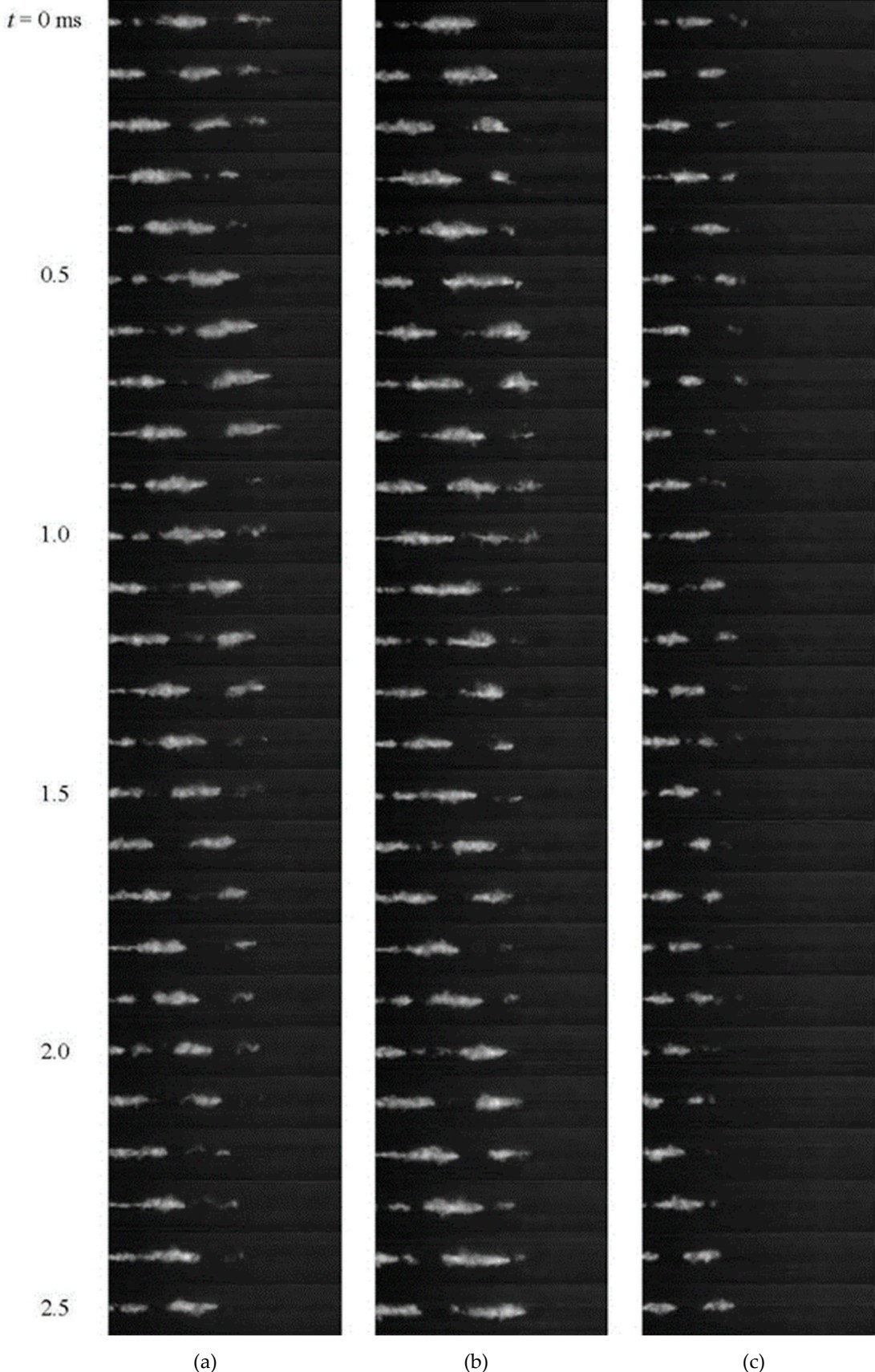

**Figure 11.** Aspects of periodical shedding of cloud cavitation: (**a**) $p_1 = 20$ MPa, $\sigma = 0.014$; (**b**) $p_1 = 30$ MPa, $\sigma = 0.014$; (**c**) $p_1 = 20$ MPa, $\sigma = 0.02$ [21].

## 4. Key Parameters of Cavitating Jet

### 4.1. Type of Cavitating Jet

As mentioned above, a cavitating jet normally means a submerged high-speed water jet with cavitation, i.e., a cavitating jet in water. Soyama developed the cavitating jet in air by injecting a high-speed water jet into a low-speed water jet without a water-filled chamber [33,35], for cavitation peening treatment outside of a tank and/or pipes [34,76]. When the residual stress of stainless steel was measured, it was reported that the cavitating jet in water introduced compressive residual stress in a deeper region, and the cavitating jet in air introduced large but shallow compressive residual stress on the surface [4]. The cavitation peening using the cavitating jet in water corresponds to shot peening using large shots, and that of the cavitating jet in air corresponds to shot peening using small shots at high velocity. Thus, the characteristics of the treated surface strongly depend on the type of the cavitating jet.

### 4.2. Standoff Distance

As shown in Figure 9b, as cavitation is generated inside and/or outside of the nozzle, it becomes cloud cavitation and forms ring vortex cavitation on the impinging surface, then collapses, producing the impacts. In view of the practical applications of the cavitating jet, the working mechanism strongly depends on the standoff distance, which is the distance from the nozzle to the surface. Figure 12 illustrates the weight loss as a function of standoff distance at constant injection pressure, $p_1$ = 120 bar (12 MPa) [77]. In Figure 12, the weight loss means a kind of aggressive intensity of the jet. Two peaks are observed at each condition in Figure 12. For convenience, the peak near the nozzle side and the peak at the further nozzle side are named the 1st peak and 2nd peak, respectively. The 1st peak results from the impacts produced by water column collisions in the jet center. Even for the submerged water jet, the similar mechanism of water jet cutting is still active, whereas the affective region is limited near the nozzle. The 2nd peak is generated by the cavitation impact, as shown in Figures 6 and 9b. As cavitation is developed and then collapsed, a certain distance from the nozzle is required. As shown in Figure 12, the optimum standoff distance $s_{opt}$ of the 1st peak is scarcely affected by cavitation number, and that of the 2nd peak strongly depends on cavitation number. At $p_2$ = 2.4 bar (0.24 MPa), the weight losses at 1st peak and 2nd peak are 250 and 450 mg, respectively. They are 250 and 220 mg for $p_2$ = 3.0 bar (0.3 MPa) and 300 and 110 mg for $p_2$ = 3.6 bar (0.36 MPa). When the maximum values of the 1st peak and 2nd peak are compared, the value of the 2nd peak at $p_2$ = 0.24 MPa is 1.5 times larger than that of the 1st peak at $p_2$ = 0.36 MPa. Namely, at optimum cavitating conditions, the aggressive intensity due to cavitation impact, i.e., the 2nd peak, is larger than that of water column impact, i.e., 1st peak. Note that water jet peening and cavitation peening use the 1st peak and 2nd peak, respectively.

To avoid confusing cavitation peening and water jet peening, Soyama proposed a classification map for cavitation peening and water jet peening using standoff distance and cavitation number, as shown in Figure 13. Over 150 points were collected from references [39,74,77–89], and it was found that the line shown in Equation (8) distinguished between cavitation peening and water jet peening [90].

$$\frac{s_{opt}}{d} = 1.8 \, \sigma^{-0.6} \tag{8}$$

One easy way to confirm the 2nd peak region, i.e., cavitating peening region, is as follows. Considering the aggressive intensity of the cavitating jet, the target materials are chosen. For example, in the case of the weak cavitating jet, a soft metal would be better. Then, the target is exposed to the jet. When a ring pattern is obtained, such as in Figure 8, it is a cavitation peening condition. The removal of paint can show the treatment area of cavitation peening [91]. A pressure-sensitive film also detects the treatment area of a cavitating jet [12].

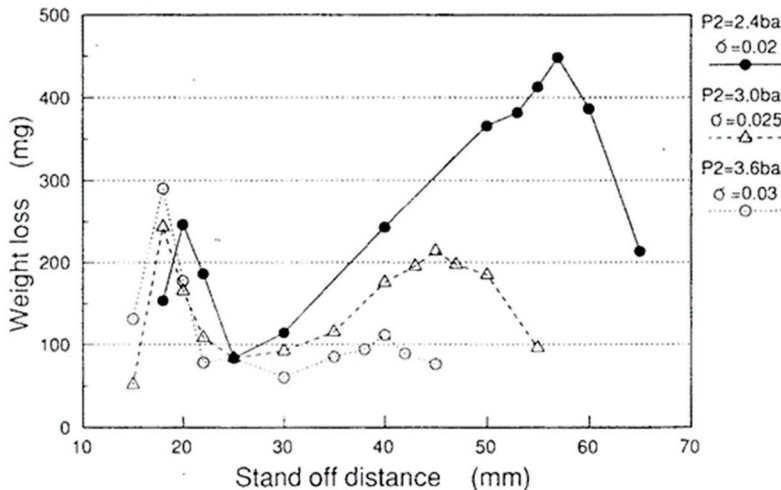

**Figure 12.** Weight loss, i.e., the aggressive intensity of the jet, as a function of standoff distance changing with cavitation number at constant injection pressure. The peak at the near side of the nozzle, i.e., the 1st peak, was caused by water column impacts. The peak at the far side from the nozzle, i.e., the 2nd peak, was produced by cavitation impacts. The standoff distance of 2nd peak was changed by the cavitation number [77].

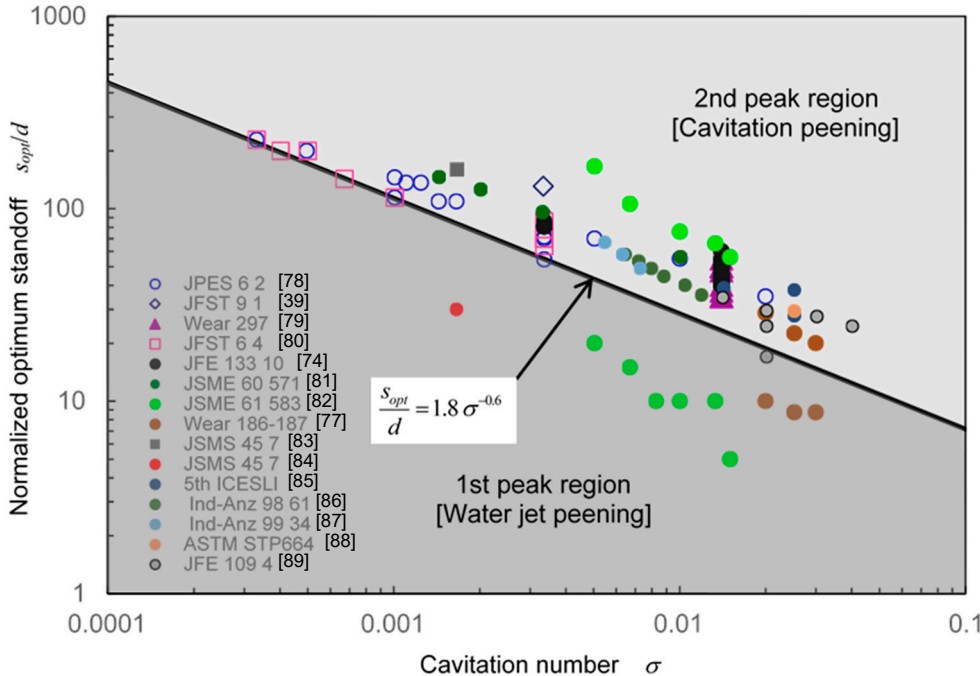

**Figure 13.** Classification map for cavitation peening and water jet peening considering standoff distance and cavitation number [90].

### 4.3. Injection Pressure

To show the effect of injection pressure on the processing capability of the cavitating jet, Figure 14 reveals the processing capability $\beta$ at (a) the 1st peak, i.e., water jet peening, and (b) 2nd peak, i.e., cavitation peening, at the constant downstream pressure condition [92]. The processing capability $\beta$ is defined by the arc height $h$ of band steel made of the same material as Almen strip, considering the width of the steel $w_s$ and the peening width $w_p$, as follows [92].

$$\alpha = \left| 1 - \frac{w_p}{w_s} \right| \tag{9}$$

$$\frac{1}{\rho} = \frac{8h}{L^2} \tag{10}$$

$$\beta = \frac{1}{\rho}(1 + \alpha) \tag{11}$$

Here, $L$ is the length to measure $h$. In Figure 14, the processing capability, i.e., a kind of aggressive intensity, was obtained by the arc height of the peened plate, as the arc height using the Almen strip is commonly used to measure the peening intensity [93]. In the case of water jet peening, $\beta$ increases with the injection pressure $p_1$, as the peening effect is produced by water column impacts, which increases with $p_1$. On the other hand, in the case of cavitation peening, $\beta$ has a maximum at $p_1 = 40$ MPa at a constant downstream condition. When the maximum values of the 1st peak and 2nd peak are compared: $\beta$ at the 2nd peak is 1.7 times larger than that at the 1st peak. As the jet power of 60 MPa is 1.8 times larger than that of 40 MPa, the peening efficiency of cavitation peening is about three times higher than that of water jet peening. Note that too high an injection pressure reduces the peening intensity of cavitation peening, as shown in Figure 14b. The reason the peening intensity of cavitation peening decreases at $p_1 > 40$ MPa is discussed in "5. Estimation of Aggressive Intensity of Cavitating Jet".

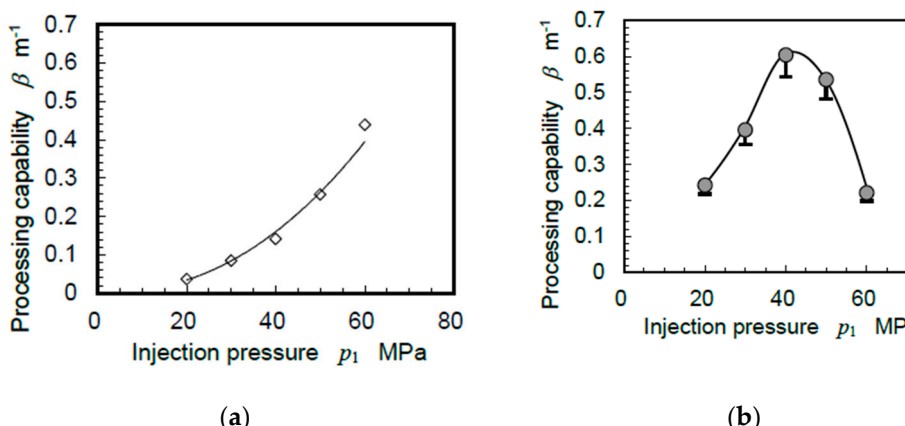

(a)             (b)

**Figure 14.** Effect of injection pressure on the aggressive intensity of cavitating jet at constant downstream pressure condition; (**a**) 1st peak; (**b**) 2nd peak [92].

### 4.4. Cavitation Number

As the cavitating jet is the cavitating flow, the cavitation number is one of the key parameters of the cavitating jet. To reveal how important the cavitation number is in the cavitating jet characteristics, the relationship between the cavitation number and the optimum standoff distance is shown in Figure 15 [72]. In Figure 15a, the cavitating length is also added, and the length and the distance are normalized by effective nozzle diameter $D_e$, which is defined by the nozzle throat diameter and discharge coefficient [72]. The data of the references are put in together in Figure 15 [77,81,86,87,89,94], as well as the relationship on each line, which is a straight line on a log–log scale. That is, the relationship can be described by Equation (12) [72].

$$\frac{s_{opt}}{d} = c_1 \sigma^{-c_2} \tag{12}$$

Here, $c_1$ and $c_2$ are constants, and they depend on the nozzle geometry.

In Figure 15b, the expected standoff distance considering the pressure distribution and the local cavitation number on the impinging surface is also shown. The expected standoff distance is on the line of the log–log scale, and it is very close to the experimental result. The local cavitation number on the impinging surface is also an important parameter when considering the flow pattern of the impinging cavitating jet.

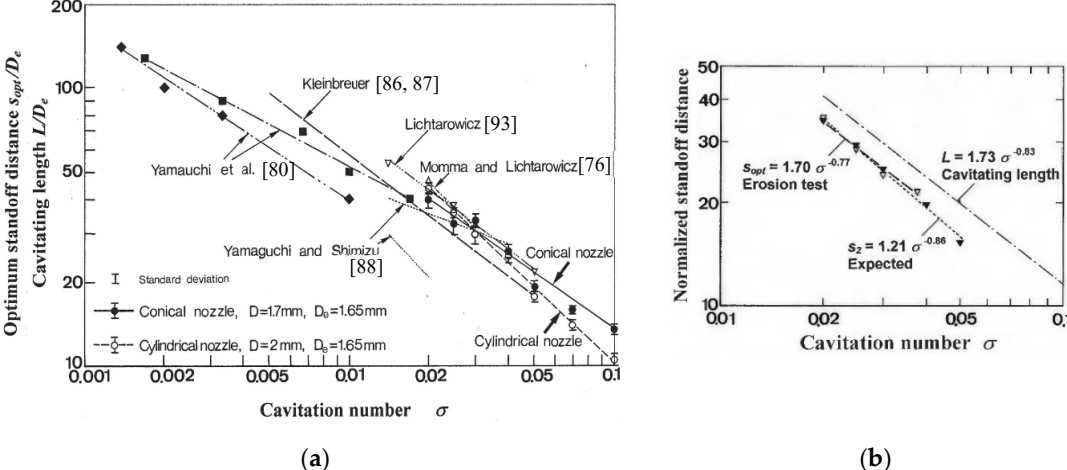

(**a**)　　　　　　　　　　　　　　　　(**b**)

**Figure 15.** Relation between cavitation number and optimum standoff distance; (**a**) experimental result of optimum standoff distance and cavitating length; (**b**) estimated standoff distance from local cavitation number [72].

### 4.5. Sound Velocity in Cavitating Flow Field

In order to consider the key factors in cavitation impacts produced by the cavitating jet experimentally, sound velocity in the cavitating flow field has been investigated [95]. As erosion rate increases with acoustic impedance [96,97], the cavitation impact might increase with sound velocity in the cavitating flow field. In view of luminescence [62,98–100] and erosion [101], the effect of sound velocity of dissolved gas has been considered; however, there is no example of evaluating the sound velocity of the cavitating flow field itself. As is well known, the sound velocity changes drastically with the void ratio [102]. In the present review, the sound velocity in the cavitating flow field is considered.

To reveal the measuring method of the sound velocity of the cavitating flow field, Figure 16 shows the aspect of the cavitating flow through a Venturi tube. In Figure 16, the water flows from the left-hand side to the right-hand side. The cavitation occurs at the throat, and the vortex cavitation is observed at the end of the cavitating region. Further downstream of the vortex cavitation, many tiny bubbles are observed: they are residual bubbles [58] after cavitation collapses. As shown in Figure 16, the densities of residual bubbles differ greatly between the left and right sides of the white arrow. When this phenomenon is observed by a high-speed video camera as shown in Figure 17, the boundary of the density, which is marked by the yellow arrow in Figure 17, moves downstream. In the experiment, the upstream pressure in the absolute pressure of the throat was 0.6 MPa, and the ratio of the cross-sectional area between the throat and the tube was 9; thus, the flow speed at the downstream of the throat was about 3.5 m/s. On the other hand, the moving speed of the boundary marked by the yellow arrow was over 600 m/s. As shown in Figure 17, at the starting point of the boundary, the vortex cavitation shrank. This aspect suggests that the pressure wave is produced by vortex cavitation collapse, and the boundary movement reveals the pressure wave, as the residual bubbles are collapsed by the pressure wave. The sound velocity in the cavitating flow field can be estimated by measuring the moving speed of the pressure wave, i.e., the boundary of the density of the residual bubbles [95].

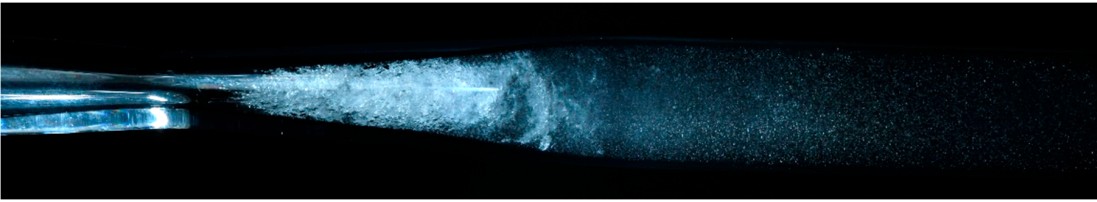

**Figure 16.** Aspects of vortex cavitation in the Venturi tube observed with a flash lamp. A pressure wave, which is indicated by the white arrow, was observed by collapsing residual bubbles.

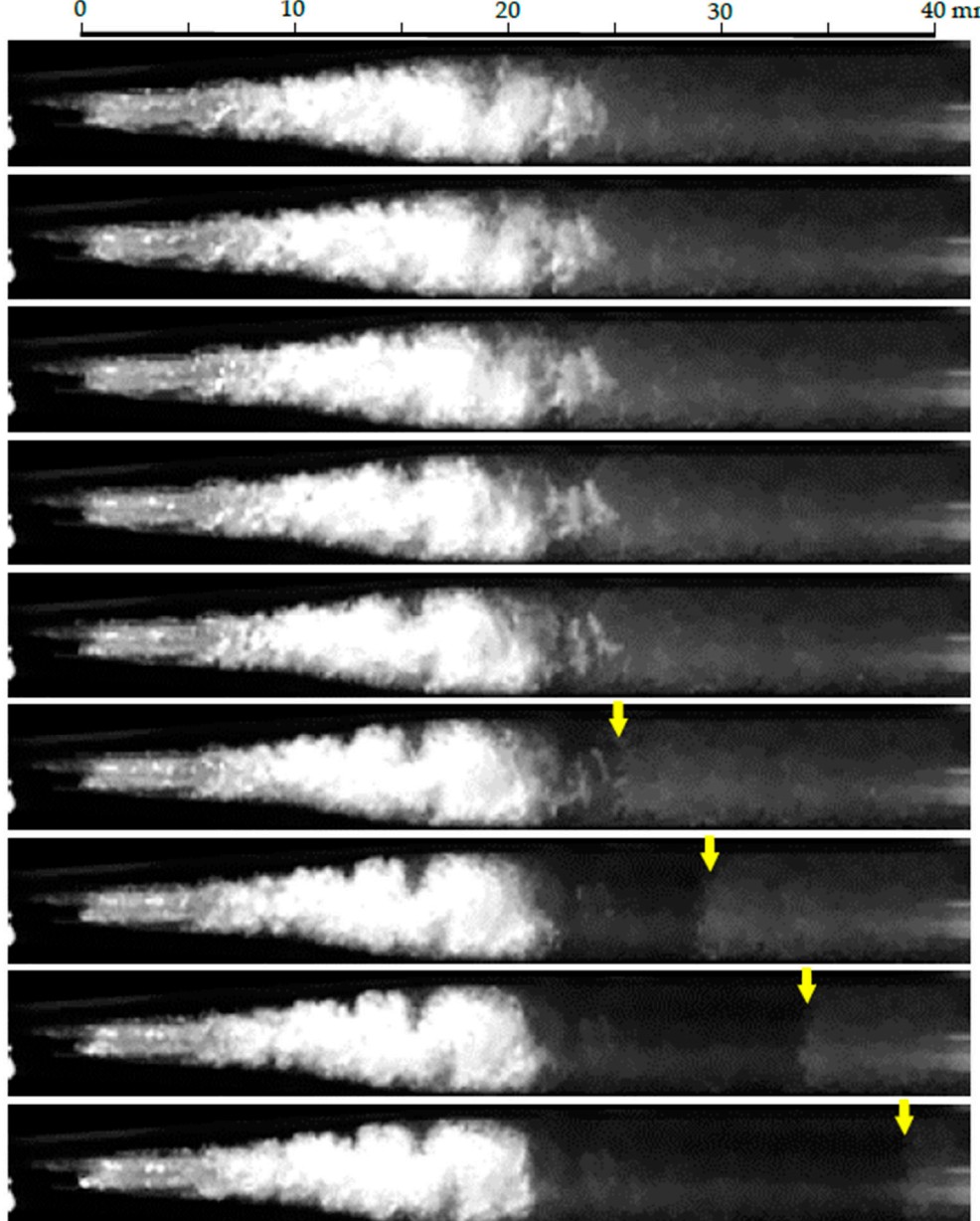

**Figure 17.** Aspects of vortex cavitation in Venturi tube observed by a high-speed video camera. The recording speed of the camera was 51,999 fps. A pressure wave, which is indicated by a yellow, arrow was observed after the vortex cavitation collapsed.

To explore the effect of cavitation number on the sound velocity, Figure 18 shows the aspect of the pressure wave changing with the cavitation number. The pressure wave is indicated by the yellow arrow in Figure 18. In Figure 18, the velocity of the pressure wave, i.e., the sound velocity, is shown in the right-hand side of the aspect. Figure 19 illustrates the relation between cavitation number and the sound velocity. The sound velocity $v_s$ increases with cavitation number. At a relatively low void ratio, the sound velocity increases with a decrease in void ratio; thus, the tendency of the relation in Figure 18 is reasonable. As mentioned above, the erosion rate increased with the acoustic impedance [96,97], and the acoustic impedance is expressed as the product of the sound velocity, the density and the speed of the microjet; thus, the sound velocity in the cavitating flow field is a parameter of the cavitating jet. This result suggests that the increase in the sound velocity is one of reasons why the aggressive intensity of the cavitating jet increases with cavitation number, as the sound velocity increases with the cavitation number, as shown in Figure 19. On the other hand, the aggressive intensity of the

cavitating jet decreases with an increase in cavitation number, as the cavitating region and/or bubble size decreases with an increase in the cavitation number. These two conflicting tendencies are the aggressive intensity of the cavitating jet has a peak for the cavitation number. More details are provided in "5. Estimation of Aggressive Intensity of Cavitating Jet". Note that the aggressive intensity of the cavitating jet had a peak at $\sigma = 0.01 - 0.014$ [4,103,104], and that of the hydrodynamic cavitation through Venturi tube had a peak at $\sigma = 0.4 - 0.7$ [99]. These peaks might depend on the flow pattern of the cavitating flow and/or the density of the residual bubbles.

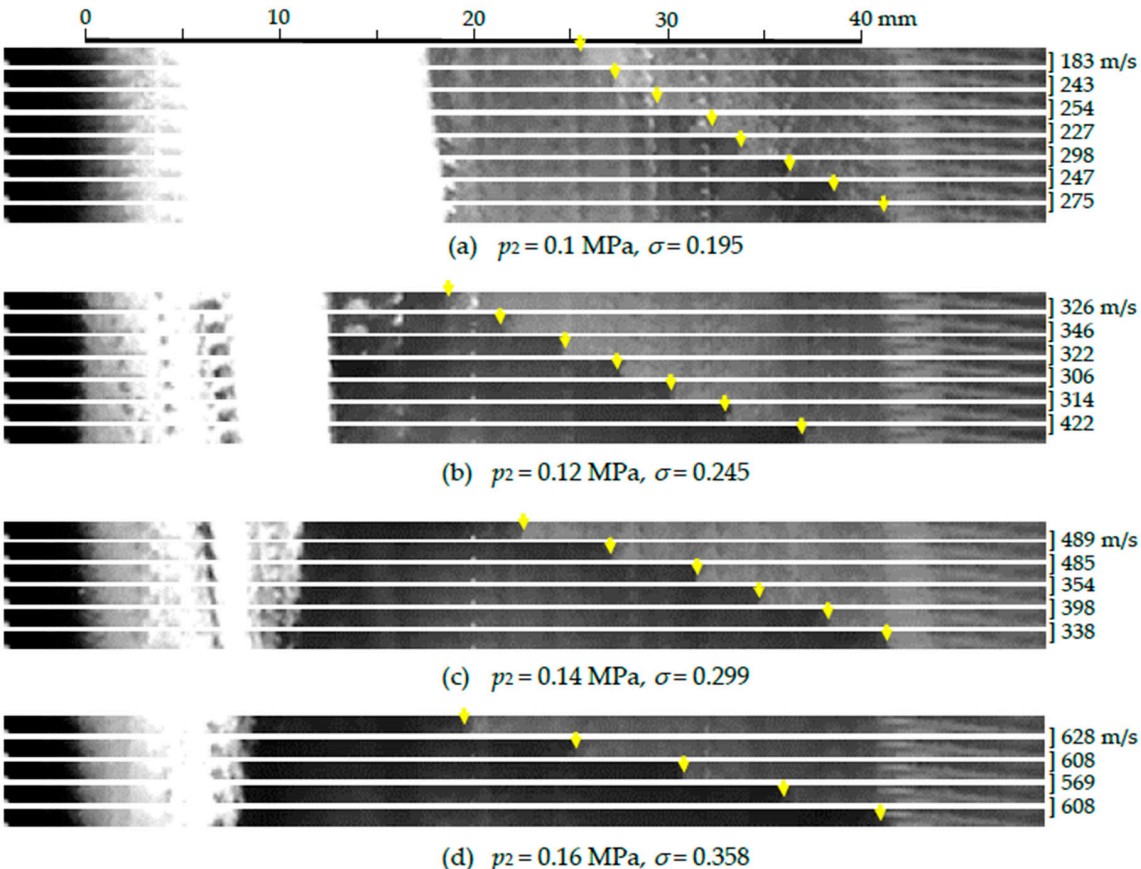

**Figure 18.** Aspects of pressure wave in Venturi tube changing with cavitation number. The recording speed of the camera was 109,999 fps.

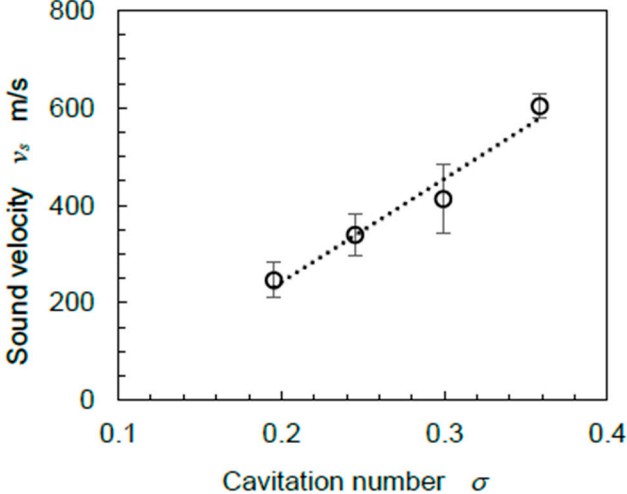

**Figure 19.** Sound velocity in cavitating flow field through Venturi tube changing with cavitation number.

### 4.6. Nozzle Geometry and Diameter

As mentioned above, the vortex cavitation in the cavitating jet is an important phenomenon, and the nozzle geometry, especially nozzle outlet geometry, is one of the key factors of the aggressive intensity of the cavitating jet, as the vortex is strongly affected by the nozzle outlet geometry. Figure 20 shows a schematic diagram of nozzle outlet geometry and the relative aggressive intensity of the cavitating jet from data published in previous reports [28,39,75,79]. Whereas a similar figure was introduced by Soyama [3], the data of nozzles K and L [28,75] were added in Figure 20. Nozzles A–E in Figure 20 are conventional nozzles for a water jet, and nozzle F is the standard nozzle for a standard test method for the erosion of solid materials by a cavitating jet [15]. Nozzle G is the nozzle obtained by optimizing the outlet bore and length experimentally [74]. Nozzle I and J had a guide pipe and a cavitator, as these enhance the aggressive intensity by about two times [39]. When both the guide pipe and the cavitator were installed, the aggressive intensity became four times larger that of nozzle J [39]. For nozzle K, when water flow holes were made near nozzle outlet, the aggressive intensity improved by 34%. When a long guide pipe with holes and water flow holes near the nozzle outlet were installed for nozzle L [28], the aggressive intensity of its cavitating jet L was 2.5 times larger than that of nozzle J and nearly 60 times larger than that of conventional water jet nozzles, as shown in Figure 20. The effect of nozzle geometry was also discussed in [14]. Resonating nozzles were also proposed and investigated [105,106].

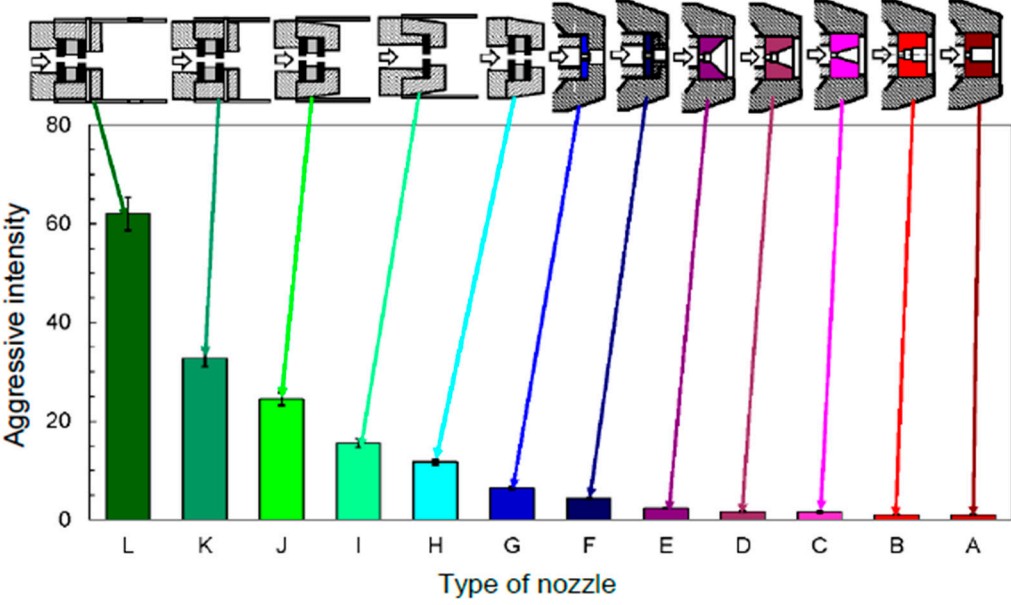

**Figure 20.** Effect of nozzle outlet geometry on aggressive intensity of cavitating jet.

The other important factor regarding nozzles is nozzle size. As mentioned above, vortex cavitation is an important phenomenon in the cavitating jet. In view of the Reynolds number, which is a key parameter of vortical flow, the larger velocity and the larger size are effective for the cavitating jet. However, too high a speed, i.e., too high an injection pressure, decreases the aggressive intensity of the cavitating jet, as shown in Figure 14b, as the sound velocity is decreased at too low a cavitation number, i.e., too-high speed condition. Thus, in the case of practical applications of the cavitating jet, the cavitating jet using a large nozzle at a relatively low injection pressure is better than that of a small nozzle at a high injection pressure [80]. The scaling law of nozzle size on the aggressive intensity of the cavitating jet is discussed in reference [107].

### 4.7. Water Qualities

As cavitation is a phase-change phenomenon from liquid phase to gas phase, as mentioned above, the water temperature affects the aggressive intensity of the cavitating jet [108–110]. It was reported that peening intensity using the cavitating jet at 288–308 K was nearly constant [4]. Whereas cavitation nuclei are required for the cavitating jet, too many air bubbles reduce the aggressive intensity of the cavitating jet due to the cushion effect [2] and the decrease in the sound velocity. When the water-filled chamber for the cavitating jet is too small, the suction vortex caused by the jet reduces the aggressive intensity of the cavitating jet. Thus, in the report, the effects of water depth and the chamber size on the aggressive intensity were investigated experimentally [111]. In the report [111], the effect of gas content on the peening intensity using the cavitating jet was also investigated using degassed water.

### 5. Estimation of Aggressive Intensity of Cavitating Jet

As mentioned above, in the constant downstream pressure condition, too high an injection pressure, i.e., too low a cavitation number, reduces the aggressive intensity of the cavitating jet. In this section, the mechanism is discussed, and a method to estimate the aggressive intensity of the cavitation jet as a function of cavitation number is proposed.

In Figure 21, the experimental results of the aggressive intensity of the cavitating jet as a function of cavitation is revealed by blue closed circles [112]. The aggressive intensity is normalized by the maximum value, and it has a peak at $\sigma = 0.016$. As the energy of cavitation is proportional to the volume of the cavitation and the pressure difference of the bubble [111], the aggressive intensity of the cavitating jet $I_{cav}$ can be assumed as follows.

$$I_{cav} \propto (L_{cav})^3 \cdot (p_2 - p_v) \tag{13}$$

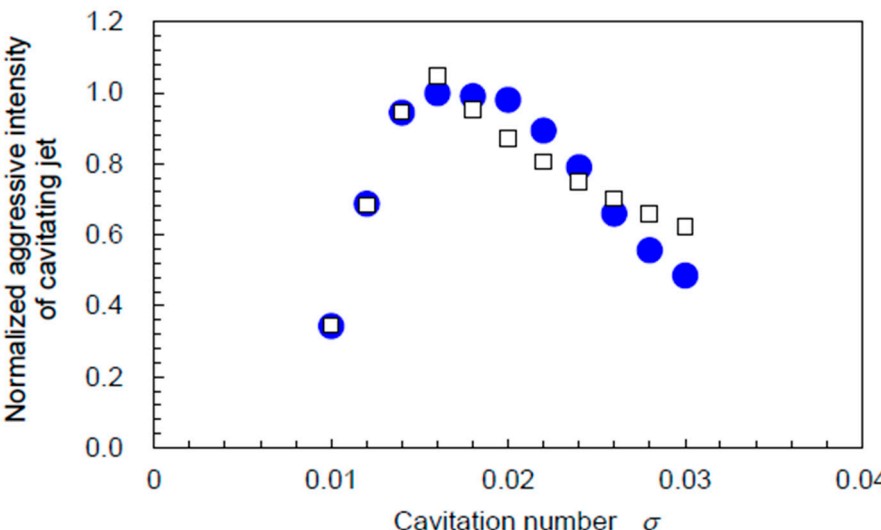

**Figure 21.** Aggressive intensity of cavitating jet changing with cavitation number.

As $I_{cav}$ is affected by the acoustic impedance [95,111], the term of the sound velocity $v_s$ is included in Equation (13). As $I_{cav}$ is also affected by flow velocity, which is defined by the pressure difference, i.e., $\sqrt{p_1 - p_2}$, the velocity term is also included in Equation (13).

$$I_{cav} \propto (L_{cav})^3 \cdot (p_2 - p_v) \cdot (v_s - v_{s\,th}) \cdot \left(\sqrt{p_1 - p_2}\right)^n \tag{14}$$

Here, $v_{s\,th}$ is the threshold level of the sound velocity considering the threshold level of $I_{cav}$ [43]. The $n$ is put as an exponent in Equation (14) to consider the power law of the velocity on $I_{cav}$ [76,105,111–113].

If it was assumed that $I_{cav}$ was proportional to the flow energy, which was defined by the product of the pressure and the flow rate, the flow energy is proportional to $p_1 - p_2$ and $\sqrt{p_1 - p_2}$, as $\sqrt{p_1 - p_2}$ was proportional to the flow velocity. Namely, the flow energy is proportional to cube of $\sqrt{p_1 - p_2}$. Thus, in the present review, $n = 3$ is chosen. From Equations (8) and (14), Equation (15) was obtained.

$$I_{cav} = c_3\, \sigma^{-1.8} \cdot (p_2 - p_v) \cdot (v_s - v_{s\,th}) \cdot \left( \sqrt{p_1 - p_2} \right)^3 \tag{15}$$

Here, $c_3$ is constant. When Equation (15) is assumed, $I_{cav}$ can be estimated by obtaining $c_3, c_4, c_5$ and $v_{s\,th}$ by a least-square method.

$$v_s = c_4\, \sigma + c_5 \tag{16}$$

The estimated $I_{cav}$ is shown in Figure 21 by empty squares. The correlation coefficient between experimental data and estimated values is 0.924. As the number of the datasets was 11, the probability of non-correlation is less than 0.005%. If the probability of a non-correlation is less than 1%, it can be concluded that the relationship is highly significant. Thus, it can be concluded that the relationship between the experimental data and the estimated values is highly significant. $I_{cav}$ can be estimated by Equation (15).

## 6. Applications of Cavitating Jet

The cavitating jet has been applied for drilling and cutting rocks [16,105,106,113]. The cavitating jet in air can also dig concrete structures for the maintenance of infrastructure [114]. In the case of mechanical surface treatment, the use of a submerged water jet was attempted to mitigate stress corrosion cracking (SCC) of nuclear power plants by using impinging impacts of a water column in the jet center at the beginning [84]. Soyama et al. found that the cavitating jet could introduce compressive residual stress into stainless steel by using cavitation impacts [13]; then, the cavitating jet was successfully applied to mitigate SCC in nuclear power plants [115]. Based on experimental results of the introduction of compressive residual stress into metallic materials, the improvement in the fatigue strength of metallic components by cavitation peening was proposed using a pressurized chamber to enhance the aggressive intensity of the cavitating jet [116–118], and it was demonstrated for forging die [119], gears [120,121], continuous valuable transmission CVT elements [122] and rollers [123]. After enhancing the aggressive intensity of the cavitating jet by optimizing the nozzle geometry, the improvement in fatigue strength by cavitation peening with an open chamber was demonstrated [17,124–127]. Cavitation peening using oil was also proposed [128]. Cavitation peening also improves tribological properties such as fretting fatigue properties [129,130]. The improvement in fatigue strength of metallic materials using a cavitating jet in air was also demonstrated [36,131,132]. In view of environmental-assisted cracking, the suppression of hydrogen-assisted fatigue crack growth in austenitic stainless steel and delayed fracture resistance on chrome molybdenum steel by cavitation peening were reported [133,134].

The cavitating jet can be applied in the semiconductor industry. It can be used not only for cleaning [135], but also gettering [136]. When erosion rates using ultrasonic vibratory apparatus [137] and cavitating jet apparatus [15] were compared, the erosion rate of the cavitating jet apparatus was larger than that of ultrasonic vibratory apparatus. As conventional ultrasonic cleaning devices use ultrasonic cavitation, cleaning using the cavitating jet is more powerful. The gettering technique is a method to remove unwanted impurities from active device regions in a silicon wafer into the back side of the wafer by introducing oxidation-induced stacking faults (OSF) [138–142]. In order to produce OSF, the introduction of strain into the wafer is required, and shot blasting is used in a conventional way [143,144]. However, the cleaning of shots is required. In the case of gettering using the cavitating jet, the cavitating jet can introduce strain for OSF [145,146] and cleaning at the same time. This is a great advantage for semiconductor processes.

One of the applications that was proposed in the bioengineering area is oral cleaning using the cavitating jet [147–149]. Dental implants have been used as the solution for the loss of teeth;

however, peri-implant mucosis and peri-implantitis are new dental diseases affecting implants [150,151]. The most effective treatment for these diseases is cleaning dental plaque, which is a kind of biofilm that adheres to the surface of teeth and implants [152]. Conventional cleaning methods of the dental plaque are oral brushing, ultrasonic scaling and rubber cup cleaning. Unfortunately, micro-textured roughness is made on the implant surface to improve biocompatibility [153], so oral toothbrushes and the tips of scaling instruments cannot reach the bottom of the rough surface. The cleaning of dental plaque on the rough surface of dental implants using the cavitating jet has been successfully demonstrated [147–149].

As is well known, sonochemistry is a research area of chemistry using ultrasonic cavitation [154]. Hydrodynamic cavitation such as the cavitating jet and cavitating flow through orifices can be applied for wastewater treatment [155–162] and to oxidize organic compounds [163]. The dispersion of spilled oil by a cavitating jet at sea has also been proposed [164]. It was reported that the efficacy of hydrodynamic cavitation was 20 times better than that of ultrasonic cavitation when the efficiency of the hydrodynamic cavitation on the pretreatment of biomass was compared with that of ultrasonic cavitation [165].

## 7. Conclusions

To use the cavitating jet for practical applications, the unsteady behavior of the cavitating jet, i.e., a submerged water jet with cavitation, was reviewed. The key factors on the aggressive intensity of the cavitating jet were also summarized. In the present review, the aggressive intensity of the cavitating jet was investigated by erosion rate and/or peening intensity. The main topics reviewed in the paper are summarized as follows.

(1) The cavitation is initiated inside and/or outside of the nozzle as a ring or helical vortex cavitation. These vortex cavitations become cloud cavitations combining with each other, and the cloud cavitation sheds periodically;

(2) Cloud shedding is a phenomenon governed by a constant Strouhal number, which is defined by the shedding frequency, the flow velocity and the width of the cavitating region;

(3) The cloud cavitation forms a ring vortex cavitation on the impinging surface and then collapses producing impacts;

(4) At optimum conditions, the affected area on the flat target by the impinging cavitating jet is a ring. The mechanism of the ring region can be explained by considering the local cavitation number on the surface. Note that the ring does not directly result from the swirl around the cavitating jet;

(5) At constant downstream pressure conditions, the aggressive intensity of the cavitating jet increases with the injection pressure, and it saturates at a certain pressure and then decreases. At too high an injection pressure, the aggressive intensity decreases.;

(6) At constant injection pressure conditions, the aggressive intensity of the cavitating jet increases with a decrease in cavitation number $\sigma$, and it saturates at $\sigma = 0.01 - 0.02$ and then decreases at too low $\sigma$;

(7) One reason why the aggressive intensity of the cavitating jet decreases at too high an injection pressure, i.e., too low a cavitation number, is the decrease in sound velocity in the cavitating flow field.

**Funding:** This research was partly supported by JSPS KAKENHI grant numbers 18KK0103 and 20H02021.

**Conflicts of Interest:** The author declares no conflict of interest.

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
