# Peer review of "Cavitating Jet: A Review"

_applsci, doi:10.3390/app10207280_

Round 1

Reviewer 1 Report

The paper provides an exhaustive review on cavitating jet. The paper is well written and it is red pleasantly. Congratulations to the authors for their work.

Author Response

Detailed Response to Reviewer 1

Comment

The paper provides an exhaustive review on cavitating jet. The paper is well written and it is red pleasantly. Congratulations to the authors for their work.

Answer

Thank you for reviewing my manuscript, and thank you for your comment.  

Reviewer 2 Report

Comment 1
Professor Hitoshi Soyama in this review thoroughly analyzes the subject of Cavitating Jet,
while giving an enjoyable reading of the text with the deposition of appropriate bibliographic references,
several of which are published work of the author,
highlighting his huge research work on this subject.
Excellent work.

Comment 2
Pages 3, 4, 6, 12, 15, 17 and 18.
Delete the space in the page end

Comment 3
Page 16 Row 354
"Figure 17. Aspects...."
Different font size from other figures.

Author Response

Detailed Response to Reviewer 2

Comment 1

Professor Hitoshi Soyama in this review thoroughly analyzes the subject of Cavitating Jet, while giving an enjoyable reading of the text with the deposition of appropriate bibliographic references, several of which are published work of the author, highlighting his huge research work on this subject. Excellent work.

Answer

Thank you for reviewing my manuscript, and thank you for your comment.  

Comment 2

Pages 3, 4, 6, 12, 15, 17 and 18. Delete the space in the page end.

Answer

Thank you for the comment. I delete the space in the page end by moving the text.  

Comment 3

Page 16 Row 354

"Figure 17. Aspects...." Different font size from other figures.

Answer

Thank you for the comment about my typo. I changed the font size.   

Reviewer 3 Report

I would recommend to check the utilization of articles within the text.

Author Response

Detailed Response to Reviewer 3

Comment

I would recommend to check the utilization of articles within the text.

Answer

Thank you for reviewing my manuscript, and thank you for your comment. I have checked the applications within the text. I added the following application in the text.

In view point of the environmental-assisted cracking, the suppression of hydrogen-assisted fatigue crack growth in austenitic stainless steel and delayed fracture resistance on chrome molybdenum steel by cavitation peening were reported [133, 134] 

  1. Takakuwa, O.; Soyama, H. Suppression of hydrogen-assisted fatigue crack growth in austenitic stainless steel by cavitation peening. International Journal of Hydrogen Energy 2012, 37, 5268-5276.
  2. Kumagai, N.; Takakuwa, O.; Soyama, H. Improvement of delayed fracture resistance on chrome molybdenum steel bolt by cavitation peening. Transactions of the JSME 2016, 82, paper No. 16-00111, 1-13.

Reviewer 4 Report

Dear Author,

Please see the original pdf file for details. Below are presented only general remarks.

First, I must admit the paper is excellent, written very concisely and clearly. I would like to add that I greatly appreciate the Author's contribution to the field of cavitation analysis. The way of description of cavitating jet phenomenon, carried out laboratory tests, and number of citations are really impressive, including the number of self-citations. It was a great pleasure to review it.

Main remarks.

Line50. ‘A typical cavitating jet in the air is shown in Fig. 1 (b).’ Fig.1b presents two cavitating jets, so why is it a typical jet? This is a combination of jets. I think the sentence should be revised.

L153. The description is not clear. Is cavitating jet shed from the nozzle continuously or stopped at 4.25s and then a new one is released? Please explain/revise the text.

L201. The Author should extend the text and explain how to understand the term 'pressure hysteresis' in case of cavitating jet? This is not obvious.

L240. What is the meaning for researchers of this statement for a good understanding of cavitation cloud shedding (CCS)? I.e., does CCS start at this number or finish? In other words, shouldn't it be inequality that CCS can be observed when St is below or above this limit (0.18)? Please explain in the text.

L269. Concerns fig.12. ‘When the maximum values of the 1st peak and 2nd peak are compared, the value of the 2nd peak is 1.5 times larger than that of the 1st peak’. In Fig. 12 such a situation can be observed only for P2 = 2.4 bar. For the others, the 1st peak is even larger. This sentence suggests such a conclusion for all three pressures shown in Fig. This should be explained.

L295. The Author should explain more clearly what it is ‘the processing capability β’ for the readers’ convenience. This is important for a better understanding Fig.14.

L439. ‘Here, vs_th is the threshold level of the velocity…’. I think the Author meant sound velocity. I suggest adding the word ‘sound’.

L440. ‘In the present review, n = 3 is chosen.’ Please explain why n = 3. This is not obvious what value should be used, so it should be clarified.

L508. ‘… a constant Strouhal number, which is defined by the shedding frequency, the flow velocity, and the cavitating region.‘ I think the last statement is not precise. Strouhal number contains the characteristic length. The Author should point directly what it is (width of jet, generally size of…, etc.).

Author Response

Thank you for reviewing my manuscript, and thank you for your comments. Please see the attached file.
